# Supervised Learning for Predictive Pore Size Classification of Regenerated Cellulose Membranes Based on Atomic Force Microscopy Measurements

**DOI:** 10.3390/ma14216724

**Published:** 2021-11-08

**Authors:** Alex Hadsell, Huong Chau, Richard Barber, Unyoung Kim, Maryam Mobed-Miremadi

**Affiliations:** 1Department of Bioengineering, Santa Clara University, Santa Clara, CA 95053, USA; ahadsell@scu.edu (A.H.); hchau@alumni.scu.edu (H.C.); ukim@scu.edu (U.K.); 2Center for Nanostructures, Santa Clara University, Santa Clara, CA 95053, USA; rbarber@scu.edu; 3Department of Physics, Santa Clara University, Santa Clara, CA 95053, USA

**Keywords:** supervised learning, atomic force microscopy, regenerated cellulose

## Abstract

Nanoporous dialysis membranes made of regenerated cellulose are used as molecular weight cutoff standards in bioseparations. In this study, mesoporous standards with Stokes’ radii (50 kDa/2.7 nm, 100 kDa/3.4 nm and 1000 kDa/7.3 nm) and overlapping skewed distributions were characterized using AFM, with the specific aim of generating pore size classifiers for biomimetic membranes using supervised learning. Gamma transformation was used prior to conducting discriminant analysis in terms of the area under the receiver operating curve (AUC) and classification accuracy (Acc). Monte Carlo simulations were run to generate datasets (*n* = 10) on which logistic regression was conducted using a constant ratio of 80:20 (measurement:algorithm training), followed by algorithm validation by WEKA. The proposed algorithm can classify the 1000 kDa vs. 100 kDa (AUC > 0.8) correctly, but discrimination is weak for the 100 kDa vs. 50 kDa (AUC < 0.7), the latter being attributed to the instrument accuracy errors below 5 nm. As indicated by the results of the cross-validation study, a test size equivalent to 70% (AUC_tapping_ = 0.8341 ± 0.0519, Acc_tapping_ = 76.8% ± 5.9%) and 80% (AUC_fluid_ = 0.7614 ± 0.0314, Acc_tfluid_ = 76.2% ± 1.0%) of the training sets for the tapping and fluid modes are needed for correct classification, resulting in predicted reduction of scan times.

## 1. Introduction

Cellulose is the most abundant biopolymer, with the common formula (C_6_H_10_O_5_)*_n_*, consisting of a linear chain of β-1,4-glycosidic bonds with varying degrees of polymerization. Cellulose-based biomaterials are used in sustainability research [1,2], nanofiber research [3,4] and tissue engineering [5,6,7], amongst other applications. Regenerated cellulose (RC) membranes are hydrophilic, solvent resistant and non-woven, with multi-scale pore sizes ranging from nm to μm. Across multiple reported porosities associated with bioseparation applications, the nanoscale pore size is used in hemodialysis [8] and bioprocessing [9,10]. Using cryopycnometry, it was shown that cyclic drying and rewetting generated a narrow skewed unimodal pore size pattern, with average pore sizes ranging from 5 to 10 nm over a range of modulated crystallinities [11]. Using scanning electron microscopy (SEM), it has been shown that RC membranes exhibited a homogenous macro- and micro-porous structure on the surface, and the inner membrane ranged from 312 to 523 nm for the surface and from 187 to 320 nm for the cross-section [12]. RC membranes characterized by a molecular weight cutoff (MWCO) of 20 kDa were used as standards for measuring the pore sizes of cross-linked alginate membranes using atomic force microscopy (AFM) [13]. 

Machine learning models that automatically generate descriptors that capture a complex representation of a material’s morphology and structure have been developed for porous material characterization [14,15,16,17]. Specifically, a random forest-based machine learning model was generated based on AFM-based roughness characterization of crosslinked nanocellulose films’ surfaces [18]. A major application of machine learning with AFM is for the correction and cleaning of AFM data [19,20,21]. Force-distance (F-D) data can be obtained from AFM scans, but indentation, baseline tilt, offset at contact point and other deviations result in rough curves. Using support vector machines and decision trees, these issues can be corrected for, resulting in smoother curves that are easier to analyze [1]. Similarly, neural networks have demonstrated the ability to increase the resolution and clarity of AFM images, allowing for easier analysis and presentation [20,21]. Algorithms have also shown an ability to remove simulated artifacts from AFM images [22]. Some convolutional neural networks are even capable of generating AFM images from scratch [21]. Using F-D curves generated by an AFM and a neural network, researchers were able to distinguish between glioblastoma and multiple myeloma cells from healthy cells [23]. Similarly, using AFM data and random forests, researchers could distinguish between cancerous and noncancerous cells found in urine with 90% accuracy [24]. The other major application of machine learning and AFM is for classification of the material being scanned by the AFM [22,23,24,25,26,27].

As captured in Table 1, over the breadth of biological samples, a critical aspect of the data used in these analyses is the resolution of the AFM. In cases where the image processing software, instrument calibration and sample fabrication are ruled out as sources of error, analysis of artifacts may be limited to spatial heterogeneity and probe aspect ratio.

The first step in biomimetic membrane design is theoretical flux calculations subject to unquantified effects, namely biofouling, concentration polarization hydrophobic interactions and electrostatic interactions. The second step is the choice of bio-fabrication method and associated materials’ interaction, often resulting in non-ideal networks, uneven shrinkage with sub-optimal pore sizes and surface pores instead of through pores, amongst other post-processing artifacts. The last step is the choice of scanning probe microscopy method resolution with regards to the sample’s intensive properties. Use of commercial RC membranes circumvents the last two challenges. Thus, the research objectives of this study are three-fold: (1) To establish a computational framework for generating pore size classifications for RC-based dialysis membranes as a model system based on discriminant analysis, as evaluated by the receiver operating characteristics curve (ROC), namely the area under the curve (AUC) and classification accuracy (Acc). (2) To assess the effect of sample size reduction for a reduction of scan time on classification quality based on a constant 80:20 training to test ratio, for the bootstrapped datasets evaluated using the holdout sub-sampling technique [28]. (3) To detect instrument accuracy errors using the above-mentioned supervised learning methodology. On the basis of this proposed algorithm, pore size calibration using certified MWCO within the range of the theoretical Stokes’ radii can be achieved. In further steps, images of synthesized membranes can be superimposed to the calibration images for discriminant analysis, with a predicted pore size range characterized by a given classification accuracy as output.

## 2. Results

### 2.1. Baseline Diffusion Measurements

Shown in Table 2 are the equilibrium absorbance differentials (A-A_0_) across MWCOs. All values exceed zero, thus confirming the hypothesis of outward diffusion of the marker from the membrane and the presence of through pores.

### 2.2. Atomic Force Microscopy

The accuracy error for the calibration standard in tapping mode was 1.4 nm (refer to Appendix A). AFM imaging was performed on single ply dialysis tubing either hydrated in saline prior to or submerged throughout imaging for the tapping and fluid modes, respectively. 

Shown in Figure 1a–d are representative 2D views with corresponding insets for multiple scan areas, displaying skewness across all samples examined. 

For the 1000 kDa, switching from tapping (Figure 1a) to fluid mode (Figure 1b) enabled capture of additional data points due to membrane hydration and improved resolution in order to focus into the deepest areas representing the smallest opening of the pores. For the 100 kDa fluid mode, hydration capture was equally beneficial, however for the 50 kDa sample, 4 μm^2^ scan areas produced blurry images (tapping mode images are not shown). These results provide a benchmark for the instrument capabilities in an attempt to correlate these findings with the classification analyses that follow.

Random sampling of pore radii across the single ply dialysis membrane resulted in 12 pores/section and 76 pores/section for the tapping and fluid modes. Combined results of the examination of instrument drift, environmental changes and statistical analysis of pore size measurements are presented in Table 3. As reflected in the absolute values of the adjusted coefficient of determination $\left(R_{adj}^{2}\right)$ deviating from unity for the filtered and unfiltered data, the effect of block variables, namely instrument drift due the spatial distribution, is negligible. The hypothetical changes in environmental conditions comprised of pore shrinkage as a result of evaporation for the tapping mode and continuous swelling for the fluid mode can also be ruled out based on the non-statistically significant *p*-values exceeding 0.05 for the Kruskal–Wallis test. The raw data for this analysis is presented in Appendix A.

For both imaging modes, wide measurement ranges could account for skewed or multimodal pore size distributions, as demonstrated in the frequency charts presented in Appendix A. Relative errors of accuracy increase with decreasing pore sizes, indicative of a lower limit of detection resolution. Notably, for the fluid mode, the average size of the 50 kDa membrane (11.1 nm) is larger than the 100 kDa membrane (8.1 nm). For the 1000 kDa sample characterized by the lowest relative accuracy measurement errors, the average size captured is 62% higher for the fluid than the one for the tapping mode. Although numerical results could show that the triangular cantilever geometry used in this study is preferable for AFM measurements in fluid due to lower drag forces [29], resonance challenges in fluid mode enhanced by the possible effect of sample tip contamination remain a source of bias between the two imaging modes. 

### 2.3. Classifier Model Development

#### 2.3.1. Normality Assessment and Hypothesis Testing for Inter-MWCO Discrimination

As presented in Table 4, all datasets failed the normality test according to the Jarque and Bera metric (*JB*) as the parameter value exceeds the critical value at the significance level of 0.05. Hence, no statistical significance test was conducted using parametric methods. 

Results of the subsequent Kruskal–Wallis test summarized in Table 5 indicate that the AFM cannot resolve the difference between the 50 kDa vs. 100 kDa membranes for either imaging mode, as reflected by the *p*-values (p>0.05). However, there is clear discrimination between 50 kDa vs. 1000 kDa (2.7 vs. 7.3 nm) and 100 kDa vs. 1000 kDa (3.4 nm vs. 7.3 nm), as well as the corresponding theoretical Stokes’ radii (p<0.05).

In the following sections, logistic regression will be conducted post non-parametric analysis for predictive modeling of pore size and the associated goal of scan reduction time.

#### 2.3.2. Algorithm Validation

Shown in Table 6 is a comparison of AUCs by membrane/imaging mode pair based on logistic regression parameters, as well as corresponding WEKA outputs for the Gamma-transformed datasets (refer to Appendix A). There is agreement between the results of the proposed methodology and the open-source software for all six combinations examined. The maximum calculated error of accuracy is 8% for 50 kDa vs. 100 kDa in tapping mode below the 10% threshold of rejection. Due the convergence of the results, the proposed algorithm has been validated against WEKA. For the 100 kDa vs. 1000 kDa combination, an acceptable AUC was obtained for both imaging modes. For the 50 kDa vs. 1000 kDa combination, discrimination was stronger for the tapping vs. fluid mode. As for the 50 kDa vs. 100 kDa discrimination, it is not possible using either imaging mode.

In conjunction with increasing measurement accuracy errors associated with smaller pore sizes previously elaborated upon based on the data presented in Table 3, cross-validation will be confined to 100 kDa vs. 1000 kDa comparisons.

#### 2.3.3. Cross-Validation and Minimum Sample Size Determination

The AUC for the datasets as well as the logistic distribution parameters obtained at the optimal Youden indices for the datasets at 100% sampling and optimal sample testing percentage, denoted by the subscript “opt”, are presented in Table 7. 

The effect of testing data size on AUC and accuracy is presented in Figure 2a,b. For both imaging modes, the standard deviation of AUC decreases with increased bootstrapped data size. Based on the optimization criteria applied to the average AUC, a test size equivalent to 70% and 80% of the training sets for the tapping and fluid modes are recommended, respectively. Retrofitting these percentages to our experimental results in terms of sample size, the test to training ratios are 34:8 and 243:61.

Furthermore, the standard deviations are an order of magnitude smaller than the average value (AUC_tapping_ = 0.8341 ± 0.0519 and AUC_fluid_ = 0.7614 ± 0.0314) for the 10 randomly generated datasets, suggesting the absence of overfitting. Corresponding accuracies are 76.8% ± 5.9% and 76.2% ± 1.0%, respectively.

Under these recommended settings, assuming a strong linear correlation between test size and scan time, the predicted AFM usage time is reduced by 30% and 20% for the tapping and fluid modes accordingly.

ROC curves resulting from 10 bootstrapped datasets generated using the optimal parameters in Table 7 as well the ones for the other comparisons for the same optimal settings (70% for tapping and 80% for fluid) are presented in Figure 3a–e. Under this reduced sampling plan, visual inspection of the curves further confirms that discriminant analysis should be confined to the 1000 kDa vs. 100 kDa.

ROC curves for the raw datasets are presented in Appendix A.

## 3. Discussion

Given the multifactorial aspect of the study encompassing environmental factors, tip contamination, instrument capabilities and non-gaussian data, the following assumptions will be made based on the results before delving into the contribution of each factor.

The pore size of the RC membrane has been assumed to be the Stokes’ radius (*r_p_*) calculated based on the MW of the manufacturer. Since the measurements have been conducted by the same individual, the magnitude of the reproducibility errors has been assumed to be negligible. As for instrument errors, periodic stage drifts have been ruled out due to poor adjusted coefficients of determination ($R_{adj}^{2}$ << 1) of Fourier fits applied to the raw datasets (Appendix A). The null hypotheses of chronological and spatial environmental shifts, specifically membrane drying for tapping and continuous membrane swelling, have been accepted for all membrane/marker combinations based on the results of the Kruskal–Wallis test (p>0.05).

Across expanding scanning probe characterization studies of cellulose nanomaterials using silicon nitride tips, encompassing contact resonance spectroscopy, chemical force spectroscopy (CFS) and atomic force microscope-based infrared spectroscopy (AFM-IR), tip contamination has been reported [30,31,32,33]. Root causes of contamination can be narrowed down to silicon residue from the PDMS storage container [34], surface adhesion of functionalized tips [30] and wear. Regarding the latter, associated root causes are chipping of metal-coated tips in Nano-IR [31,32] and tip contamination due to sample wear [33]. In the current study, precautions were taken to change the tip before each scan in order to minimize sample wear residue, however the tips were not cleaned prior to usage. Possible sources of contamination are silicon oil residue and dialysis membrane wear, resulting in decreased resolution for both imaging modes. For the tapping mode, the interaction volume between the probe and the surface is made larger by the contamination layer, while for the fluid mode, the balance of hydrodynamic forces may be disturbed by micelle formation or random residue deposits.

In the absence of statistically significant spatial and temporal shifts throughout the scan (Appendix A) along the use of non-functionalized and uncoated tips, it has been assumed that the unquantified effect of contamination is a source of systematic error.

Cellulose-based hydrogels are susceptible to swelling [35], with a reported swelling ratio (SR) ranging from 50% to 3000% for regenerated cellulose [36,37]. A range of SR = 50–200% has been adapted to calculate the theoretical pore radius in order to interpret the departures from the manufactured values. The objective is to find an initial swelling ratio range based on a common direction of change for all three standards for measurement error minimization, assuming that the swelling rate is the same at all MWCOs. Shown in Figure 4 are the absolute errors of measurement accuracy adjusted to swelling ratios (SR) calculated based on the theoretical Stokes’ radii. Between the ratios of 50–100%, accuracy errors are decreasing in the same direction for all RC membranes. Specifically, for the 1000 and 100 kDa membranes, the error is within ±1.5 nm for SR = 75–100%, in contrast to the 50 kDa membrane with approximately double the error range (6–7 nm). For the latter, accuracy is inversely proportional to the swelling ratio, and an optimum of 3 nm is reached at SR = 200%. At this maximal projected swollen state, the 1000 kDa measurement error is maximal, contradicting how the measured relative errors of accuracy increase with decreasing pore sizes for the fluid mode. Thus, initial hydrogel swelling state cannot account for the poor instrument resolution between the 100 and 50 kDa RC membranes.

A critical aspect of the data used in these analyses is the spatial resolution of the AFM, in this study estimated to be ±1.5 nm using the calibration step in tapping mode (refer to Appendix A). Uncertainties in these measurements can have a significant impact on the discriminant analysis, especially when the smallest features are considered. In this study, the smallest error of accuracy of 2 nm is within the uncertainty range of the calibration step measured in tapping mode for the 1000 kDa RC membrane. Results of another study conducted in tapping mode on the same membrane feature—extracted at 0.25 μm^2^ instead of 1 μm^2^—resulted in pore diameters ranging from 13 to 18 nm with no specified sample size, as compared to a range of 5–28 nm [38]. Theoretically, a decrease in scan area allows the ability to zoom into the deepest areas of the image, which represents the smallest opening of the pores assuming that the resolution of the scanner is adequate.

In order to determine the lateral resolution of an AFM, Moiseev et al. calculated the contribution to the van der Waals forces for various tip radii and tip-to-sample distances under ambient conditions. They found that for a tip radius of curvature of 5 nm, lateral resolution could approach 2 nm if the tip was held very close (0.5 nm) to the surface [39]. A more detailed analysis by Gan concludes that even larger working distances can be impractical due to attractive forces pulling the tip to the surface. The main conclusion regarding lateral resolution is that the theoretical limit is strongly determined by the tip geometry [40]. Furthermore, given the instrument- and environment-dependent nature of AFM measurements, it is perhaps more fruitful to consider resolution results from experimental studies for tips with a radius of curvature of 10 nm, typical of the NanoWorld PNP-TR tips utilized in this work [13,41], for which the best lateral resolution ranged between about 3 and 5 nm. The most accurate result has been obtained for an RC membrane with a MWCO of 20 kDa ($r_{p}$ = 2.02 nm). Using the tapping mode, an average pore diameter of 4.9 nm was measured corresponding to an accuracy error of 143%, consistent with the obtained results for Stokes’ radii below 5 nm, as presented in Table 3 [13]. In another study, a direct resolution comparison was performed experimentally with double-wall carbon nanotubes (DWNTs) and a conventional 8 nm radius of curvature Si tips. The results indicated lateral resolutions of 5–6 nm for the DWNTs and 32–35 nm with the Si tips [42]. These results further support that the observations of the AFM instrument used in this work could not reliably resolve features below 5 nm. AFM imaging in fluids using tapping mode is subject to resonance challenges. In water, the relatively low Q-factor (20–30) of the cantilever results in a degradation of spatial resolution [43]. A method for overcoming this challenge is frequency modulated AFM (FM-AFM) conducted in fluid and tapping modes [44]. Tip treatments can also improve the interaction of the tip with the sample, and atomic resolution is possible. Comparisons between untreated and treated tips show a dramatic resolution enhancement, with untreated tips producing lower quality images [42].

In this study, a sample size imbalance occurred between the two imaging modes, where 60 points vs. 304 points were sampled for the tapping and fluid modes, respectively. The lower sample size for the former was to prevent sample shrinkage at lower scan rates, while the higher sample sizes were designed to increase the power of the test, the latter being susceptible to resonance challenges [43]. Nevertheless, each dataset contained a minimum of 30 samples, maximizing the chances of the Central Limit Theorem to hold in the absence of skew.

Regardless of the fabrication and characterization method, the pore size distribution of RC membranes has been reported to be skewed, in agreement with the results of this study [10,19,37]. As shown in Table 8, there is agreement between AUCs generated from the raw datasets (refer to Appendix A) and the those obtained from proposed logistic regression models at 100% sampling, except for the combination F2. In future studies with enhanced instrumentation capability, outlier analysis will be conducted to further examine the extent of the inherent sample skewness incorporated into algorithm development.

## 4. Materials and Methods

### 4.1. Materials

Spectrum™ Labs Regenerated Cellulose (RC) dialysis membranes with molecular weight cutoffs (MWCO)s of 50 kDa (08-700-128), 100 kDa (08-700-132) and 1000 kDa (08-801-255) were purchased from Fisher Scientific (Waltham, MA, USA). The tri-angular Pyrex-Nitride AFM probes (PNP-TR-20) were purchased from NanoWorld (Neuchâtel, Switzerland). The AFM Calibration block (TGZ1) characterized by a step height of 20 ± 1.5 nm was purchased from Ted Pella Inc. (Redding, CA, USA). Fluorescein isothiocyanate dextran MW markers, 4 kDa (FD), as well reagent grade salts were purchased from Sigma Aldrich (Saint Louis, MO, USA). 

### 4.2. Methods

#### 4.2.1. Baseline Diffusion Measurements

Baseline diffusion measurements were carried out to ensure porosity of the synthetic membranes prior to AFM measurements. Calibration stock solutions of 5 mg/mL for each FITC-Dextran MW standard (4 kDa or Stokes’ radii of 1.4 nm) dissolved in 0.9% (*w*/*v*) NaCl were prepared. In dilute solutions, there is a linear relationship between absorbance and the concentration of the marker under observation, thus diffusion measurements were carried out at this nominal concentration. Dialysis tubing of various MW cutoffs were filled with 2 mL of MW marker solutions and subsequently incubated and stirred at 120 rpm in a 10 mL beaker filled with 0.9% (*w*/*v*) NaCl. The supernatant absorbance was measured prior to (A_0_) and post-mixing at 350 nm using a Genesis 10 S UV-Vis spectrophotometer (Thermo Fisher Scientific, Waltham, MA, USA). The absorbance value at 8 h (A) was taken to be the equilibrium value.

The molecular weight cutoff (MWCO) of the membrane expressed in terms of Stokes’ radius ($r_{p}$) was given by Equation (1) [45]. This equation assumes that the solute of molecular weight (MW) is a sphere with a density (*ρ* = 1 g·cm^−3^) equal to that of the solute in solid phase.
(1)$$r_{p}\;=\;{\left(\frac{3M_{w}}{4\rho \pi N_{A}}\right)}^{1/3}$$

#### 4.2.2. Atomic Force Microscopy

Imaging was performed via a 3100 Dimension atomic force microscopy machine (DAFM-XYZ, Bruker Instruments, Billerica, MA, USA). The atomic force microscopy (AFM) scan was conducted in tapping and fluid modes using a Pyrex-Nitride probe (PNP-TR-20, NanoWorld, Neuchâtel, Switzerland) with a triangular cantilever (resonant frequency 17 kHz, force constant 0.08 N/m, thickness 500 nm, length 200 μm, tip radius 7–10 nm). A total of 6 tips were used, corresponding to the respective MW marker/imaging mode outlined in Table 6. Nanoscope v6.13 (Bruker Instruments, Billerica, MA, USA) and Gwyddion v2.3 (Czech Metrology Institute, Brno, Czechoslovakia) were used as image analysis software, respectively.

Scan speed was established by setting a ratio of 512 pixels/line for a range of frequency of 0.1–0.2 Hz. The scan area ranged from 1 to 8.0 μm^2^. The pore sizes were feature-extracted at a constant scan area of 1 μm^2^. Images were obtained at 5 different locations for the tapping and 4 different locations for the fluid modes respectively, across the sample. 

The dialysis tubing, stored in sodium azide to avoid biofouling, was cut in order to produce a 1 cm^2^ single ply sheath. For the tapping mode, samples were washed and allowed to equilibrate in filtered saline (0.9% *w/v* NaCl) for 30 min. For the fluid mode, the sample remained submerged in the saline throughout the scan.

#### 4.2.3. Detection of Instrument and Environmental Drifts

Feature-extracted pore size measurements were fitted to Fourier transforms by using the nonlinear least-squares formulation of Matlab2020a with and without robust weighing options. The resultant output in terms of the adjusted coefficient of determination ($R_{adj}^{2}$) was used to detect periodicity in data caused by hypothetical instrument drifts during imaging for the sections under observation.

Changes in environmental conditions by sample scan location were assessed by the non-parametric Kruskal–Wallis test, equivalent to a one-way analysis of variance (ANOVA) for non-normally distributed data using the Matlab2020a kruskalwallis function at the 95% confidence interval (CI) [46].

#### 4.2.4. Classifier Model Development

##### Normality Assessment and Hypothesis Testing for Inter MWCO Discrimination

Normality was quantitatively assessed using the Jarque-Bera test [47] combining the central distribution moments given by Equation (2):(2)$$JB\;=\;n\left(\frac{{\left(\sqrt{b_{1}}\right)}^{2}}{6}+\frac{{\left(b_{2}-3\right)}^{2}}{24}\right)$$
where *b*_1_ and *b*_2_ are the sample skewness and sample excess kurtosis, *n* is the sample size and *JB* is the Jarque-Bera metric, respectively. The *JB* test statistic is approximately Chi-squared (χ^2^)-distributed, under the assumption that the null hypothesis is true. It is equal to zero when the distribution has zero skewness and kurtosis is 3. 

Hypothesis testing using parametric and non-parametric methods was carried out to assess whether the RC membranes may be distinguished based on mean/median values at the 95% CI. If the data were normally distributed, hypothesis testing was used, using a two-sample t-test with the Matlab2020a ttest2 function. Homoscedasticity was conducted using the Levene test (Matlab2020a vartestn) function prior to conducting the two-way comparisons at the 95% CI to compare the mean pore size distributions [46]. If the data were not normally distributed, the Kruskal–Wallis test was used at the 95% CI.

##### Data Transformation

Pending negative normality, the data were fitted to the Gamma distribution using the Matlab2020a gamfit function, in order to address the skewness of the raw data. Random variable *X* has a Gamma distribution if its probability distribution function is given by Equation (3), where *a* and *b* (*a >* 0 and *b >* 0) are respectively the shape parameter and scale parameter [46]:(3)$$f\left(x\right)\;=\;\frac{1\;}{b^{a}\Gamma \left(a\right)}x^{a-1}e^{-\frac{x}{b}},\;\;\;x\geq 0$$

##### Logistic Regression and Receiver Operating Characteristic Curves

Following distribution parameter determination for the Gamma distribution, a Monte Carlo simulation was used to generate 10 datasets for each membrane type/imaging mode pair. Logistic regression (Equation (4)) was conducted on the datasets using a constant 80% of the data for training and 20% for testing, for the randomly generated datasets. The binomial link function is provided between the probability, *p*, and the covariate, *x*, for a binomially distributed variable:(4)$$logit\left(p_{i}\right)\;=\;log\frac{p_{i}\;}{1-p_{i}}\;=\;\beta_{0}+\beta_{1}x_{i},\;\;i\;=\;1,\ldots ,k,\;\overset{k}{\underset{i=1}{\sum }}n_{i}\;=\;n$$

ROC curves were generated using the logistic link function. Discrimination strength in terms of the area under the curve (AUC) was evaluated qualitatively using the categorical scale presented in Table 9. 

A validation step was conducted by comparing the output of the software WEKA (Waikato Environment for Knowledge Analysis, University of Waikato, New Zealand) in terms of relative error of AUCs with an acceptable limit of 10% and below.

##### Cross-Validation and Optimal Testing Sample Size Determination 

Following algorithm validation, the size of the bootstrapped training set was subsequently varied from 10% to 100% according to the sample hold-out method in order to assess the effect of sample size reduction on AUC. The two-fold optimization criteria for the size are a minimal rate of change for the average value and an AUC > 0.7 after subtracting the standard deviation from the mean value.

The logistic regression parameters evaluated at the optimal Youden’s index ($Y_{i}$) and the corresponding classification accuracy values are reported. The Youden’s index was calculated using Equation (5), where$\;\left(Se_{i}\right)$ and $\left(Sp_{i}\right)\;$are the sensitivity and specificity at each cutoff, respectively [46]:(5)$$\;\;\;\;\;\;\;\;Y_{i}=max\left(\frac{Se_{i}+Sp_{i}-1\;}{\sqrt{2}}\right)$$

Classification accuracy (Acc), given by Equation (6), is defined as the rate of correct classification, where (TP) and (TN) are the true positives and true negatives respectively, and (N) is total number of classified samples.
(6)$$Acc\;=\;\left(\frac{TP+TN\;}{N}\right)\;\;\;\;\;\;\;\;\;\;\;\;\;\;\;\;\;\;$$

## 5. Conclusions

Using supervised learning, the specific aims of pore size classification and reduction of scanning time by predictive modeling have been met. Use of RC membrane standards enabled detection of instrument accuracy errors associated with the AFM spatial resolution, thus enhancing the classification quality. Higher measurement accuracies are achieved in tapping mode. Resonance challenges created by pure hydrodynamic forces need to be decoupled from those added from sample tip contamination in order to quantify the bias between the average pore sizes reported for each method. Results of non-parametric analyses and logistic regression models coupled to projected hydrogel swelling analysis point to a loss of instrument resolution below 5 nm. With a hypothesized strong correlation of test size to scan time, in the future, researchers may shorten the experiment time for the fluid and tapping modes accordingly, and dedicate the saved time to replication. Future efforts will explore the use of FM-AFM capabilities, tip treatment and a dew point controller in order to improve the lower limit of detection, followed by the subsequent validation using the proposed classifier algorithms. Of particular interest is to establish a framework for the predictive analysis of pore sizes obtained using scanning probe microscopy from biomimetic anisotropic materials [48,49]. Under enhanced capture methods for improving spatial resolution, parallel supervised learning techniques namely support vector machines will be run in parallel and cross-validated to compare the extent of overfitting as well as examining the nature of the outliers contributing to non-Gaussian data [28].

## Figures and Tables

**Figure 1 materials-14-06724-f001:**
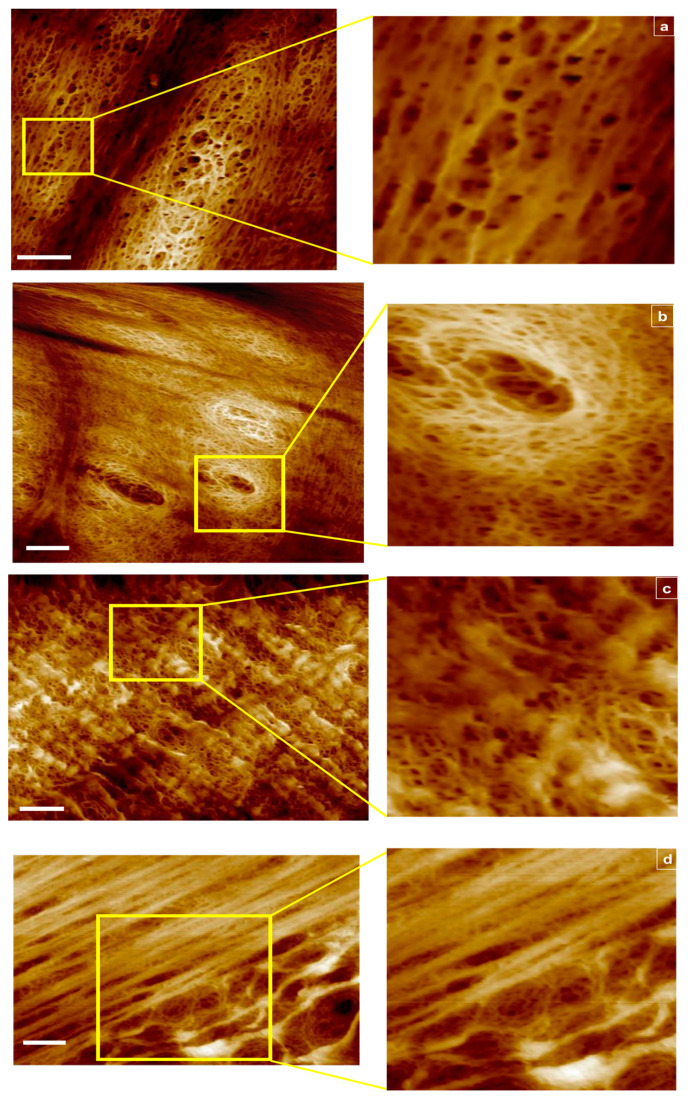
AFM images of RC membranes from top to bottom: (**a**) 1000 kDa in tapping mode scan area 6 μm^2^, 2 μm^2^ section magnified, (**b**) 1000 kDa in fluid mode, scan area 8 μm^2^, 2 μm^2^ section magnified, (**c**) 100 kDa in fluid mode scan area 8 μm^2^, 2 μm^2^ section magnified, and (**d**) 50 kDa in fluid mode scan area 8 μm^2^, 4 μm^2^ section magnified. Scale bar represents 1 μm.

**Figure 2 materials-14-06724-f002:**
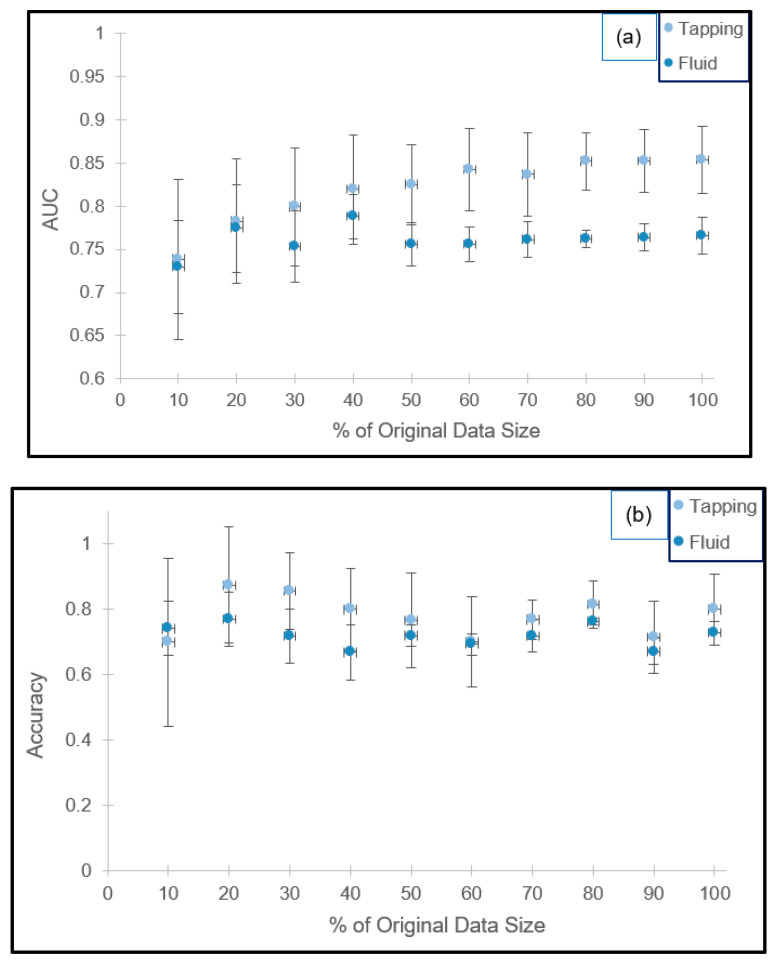
Cross-validation results for the 100 vs. 1000 kDa combinations in tapping and fluid modes in terms of the AUC ((**a**) top) and classification accuracy ((**b**) bottom).

**Figure 3 materials-14-06724-f003:**
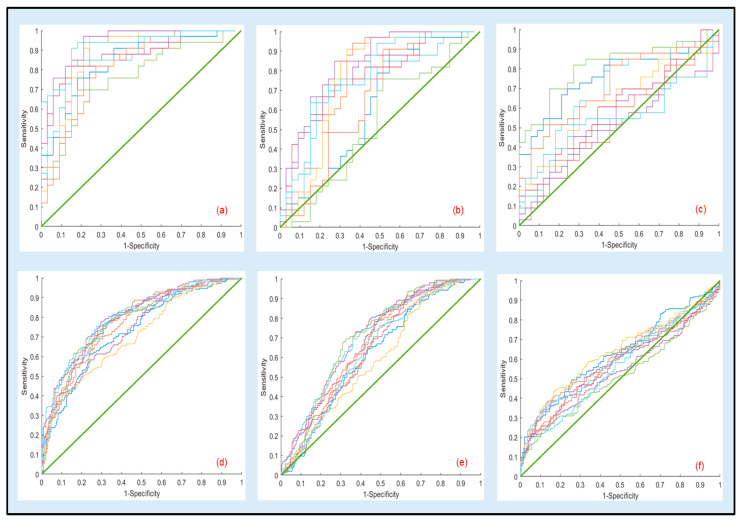
Representative ROC curves for the cross-validation results generated using logistic regression parameters from 10 bootstrapped datasets estimated at the optimal testing sizes of 70% for tapping and 80% for fluid. Top and bottom rows correspond to tapping and fluid modes, respectively: (**a**,**d**) 100 kDa vs. 1000 kDa in tapping and fluid modes, (**b**,**e**) 50 kDa vs. 1000 kDa in tapping and fluid modes, (**c**,**f**) 50 kDa vs. 100 kDa in tapping and fluid modes. Regression parameters for (**a**,**c**) are presented in Table 7.

**Figure 4 materials-14-06724-f004:**
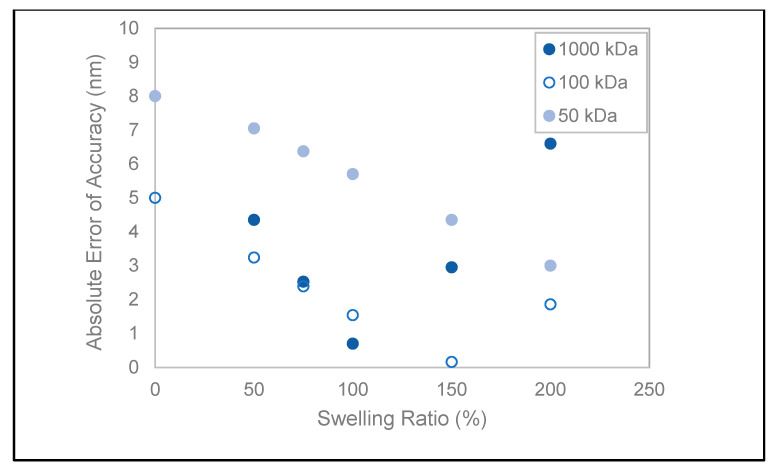
Absolute measurement accuracy error as a function of the hypothetical swelling ratio.

**Table 1 materials-14-06724-t001:** Materials characterized using AFM and machine learning.

Material	Machine Learning Method(s)	Study
Zebrafish spinal cord	Support vector machines Decision trees	Müller, et al. [19]
Glioblastoma and multiple myeloma cells	Neural network	Minelli, et al. [23]
Cells found in urine	Random forests, extremely randomized forests, gradient boosting trees	Sokolov, et al. [24]
Ferritic electric materials and electrochemical systems	Support vector machines	Huang, et al. [25]
Ionic liquid layers on top of graphite and melem on boron nitride	Fast-Fourier Transform (FFT), primary component analysis (PCA), cross-correlation (CC) and stationary wavelet decomposition	Borodinov, et al. [26]
DNA fragments	Linear regression	Sundstrom, et al. [27]
Human leukemia cells AFM images with simulated artifacts	Support vector machines and linear discriminant analysis	Mencattini, et al. [22]
Low-resolution AFM images	Convolutional Neural Network (CNN)	Liu, et al. [20]
Simulated AFM images	Convolutional Neural Network (CNN)	Oinonen [21]

**Table 2 materials-14-06724-t002:** Diffusion results evaluated by spectrophotometry.

Marker/Membrane MWCO	(A-A_0_)
4 kDa/50 kDa	0.37
4 kDa/100 kDa	0.61
4 kDa/1000 kDa	1.13

**Table 3 materials-14-06724-t003:** Pore size measurements and associated errors of measurement accuracy.

Statistic	Mode	n	1000 kDa	100 kDa	50 kDa
Theoretical Stokes’ radius (nm)	N/A	N/A	7.3	3.4	2.7
Average (nm)	Tapping	60	9.45	7.34	5.65
Measured range (nm)	Tapping	60	23	41	19
% relative error of accuracy	Tapping	60	29.4	116	109
absolute error of accuracy (nm)	Tapping	60	2	4	3
$R_{\mathrm{adj}}^{2}$ Fourier fit raw data	Tapping	60	0.430	0.091	0.200
$R_{\mathrm{adj}}^{2}\;$Fourier fit filtered data	Tapping	N/A	0.290	0.212	0.380
Kruskal–Wallis test p-value	Tapping	60	0.5384	0.9569	0.924
Average (nm)	Fluid	304	15.3	8.34	11.1
Measured range (nm)	Fluid	304	138	44	72
% relative error of accuracy	Fluid	304	110	145	312
absolute error of accuracy (nm)	Fluid	304	8	5	8
$R_{\mathrm{adj}}^{2}$ Fourier fit raw data	Fluid	304	0.057	0.036	0.02
$R_{\mathrm{adj}}^{2}$ Fourier fit filtered data	Fluid	N/A	0.398	0.371	0.450
Kruskal–Wallis test p-value	Fluid	304	0.1534	0.681	0.681

**Table 4 materials-14-06724-t004:** Summary of normality assessment conducted on raw datasets.

RC Type	Gaussian Moments	Tapping (*n* = 60)	*JB **	Normality Assessment	Fluid (*n* = 304)	*JB ***	Normality Assessment
50 kDa	Skewness	2.069	113	Fail	2.307	733	Fail
	Kurtosis	8.201			9.048		
100 kDa	Skewness	2.599	371	Fail	2.833	1545	Fail
	Kurtosis	9.338			12.482		
1000 kDa	Skewness	2.314	117	Fail	4.776	17,919	Fail
	Kurtosis	8.017			39.379		

* *JB_critical_* = 5.1; ** *JB_critical_* = 5.7.

**Table 5 materials-14-06724-t005:** Hypothesis testing results for the Kruskal–Wallis test at the 95% confidence interval.

Mode	Comparison	*p*-Value
Tapping	100 vs. 1000 kDa	0.000
Tapping	50 vs. 1000 kDa	0.000
Tapping	50 vs. 100 kDa	0.275
Fluid	100 vs. 1000 kDa	0.000
Fluid	50 vs. 1000 kDa	0.000
Fluid	50 vs. 100 kDa	0.083

**Table 6 materials-14-06724-t006:** Comparison of algorithm-generated AUCs and WEKA outputs by imaging mode for a split ratio of 80:20 (training: test).

Mode	Comparison	Weka AUC (80%)	Algorithm AUC (80%)
Tapping	100 vs. 1000 kDa	0.931	0.852 ± 0.0335
Tapping	50 vs. 1000 kDa	0.785	0.722 ± 0.0424
Tapping	50 vs. 100 kDa	0.590	0.604 ± 0.0437
Fluid	100 vs. 1000 kDa	0.753	0.762 ± 0.0104
Fluid	50 vs. 1000 kDa	0.663	0.661 ± 0.0184
Fluid	50 vs. 100 kDa	0.576	0.579 ± 0.0172

**Table 7 materials-14-06724-t007:** Comparison of classifier strength at 70% for tapping mode and 80% for fluid mode.

Method	Comparison	AUC at 100%	AUC_opt_	Youden at 100%	Youden_opt_	β_0opt_	β_1opt_
Tapping	100 vs. 1000 kDa	0.8521 ± 0.0335	0.8341 ± 0.0519	0.442 ± 0.0496	0.463 ± 0.0582	−4.1233 ± 0.9533	0.5998 ± 0.1316
Fluid	100 vs. 1000 kDa	0.7622 ± 0.0104	0.7614 ± 0.0314	0.287 ± 0.0209	0.294 ± 0.0438	−1.7784 ± 0.2600	0.1561 ± 0.0256

**Table 8 materials-14-06724-t008:** Comparison of classifier agreement between raw datasets and the proposed algorithm.

Combination	Comparison	Raw Data AUC	Algorithm AUC (100%)	Agreement
T1	100 kDa vs. 1000 kDa	0.894	0.8521 ± 0.0335	Yes
T2	50 kDa vs. 1000 kDa	0.749	0.7216 ± 0.0425	Yes
T3	50 kDa vs. 100 kDa	0.620	0.6035 ± 0.04366	Yes
F1	100 kDa vs. 1000 kDa	0.794	0.7622 ± 0.0104	Yes
F2	50 kDa vs. 1000 kDa	0.708	0.6613 ± 0.0184	No
F3	50 kDa vs. 100 kDa	0.541	0.5821 ± 0.0385	Yes

**Table 9 materials-14-06724-t009:** Categorical scale for AUC.

Strength	AUC
high	0.8–1
medium	0.7–0.8
low	<0.7

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
