# Peer review of "Supervised Learning for Predictive Pore Size Classification of Regenerated Cellulose Membranes Based on Atomic Force Microscopy Measurements"

_materials, 2021, doi:10.3390/ma14216724_

Round 1
Reviewer 1 Report
I have read this manuscript many times and tried to find its value for pore size prediction. However, it is very hard to find the actual significance. So, I make the difficult decision to give the suggestion about reject.
Author Response
Dear Reviewer,
Thank you for your valuable comment.
The last paragraph of the introduction has been expanded to explain the rationale behind conducting the study.
I hope that this clarifies the specific aims more clearly.

Reviewer 2 Report
The authors investigated the possibility for automated classification of the pose size in regenerated cellulose membranes. As an experimental method to get the images of the pores in the membrane they used atomic force microscopy. Then the authors applied special machine learning models to perform the pore size classification. The proposed method could have an important practical application not only in the field of cellulose membranes but also in other porous materials.
I have the following suggestions for a minor revision:
- Table 3 in the column statistics: What means Average, is the measured diameter of the pore? Why the average for the fluid mode and for the taping mode is so different?
- The introduction and eventually Table 1 could be extended with an application of the machine learning methods in the field of peptide design. A recent paper using a similar bioinformatics approach that can be appended to the list of references is "Prediction of Amphiphilic Cell-Penetrating Peptide Building Blocks from Protein-Derived Amino Acid Sequences for Engineering of Drug Delivery Nanoassemblies", Journal of Physical Chemistry B, 2020, 124(20), pp. 4069-4078.
Author Response
Dear Reviewer,
Thank you for your valuable comments.
Regarding comment #1, the differences in pore size between both imaging modes, responses are found in the results and discussion section. The conclusion addresses the need for investigating this bias that could be due to a combination of contamination and resonance challenges.
Regarding comment #2, the reference has relevant has been added.
Please find attached a document with more details.

Reviewer 3 Report
In their manuscript "Supervised Learning for predictive pore size classification ...", Hadsell and coworkers report on a machine learning approach to extract some relevant quantities from AFM images. Machine learning based image analysis will become more and more relevant in the future, especially in cases where a large number of images have to be analyzed. This definitely makes the paper relevant for publication in "materials", even so the paper is rather difficult to read for non-experts in machine learning. Therefore, the paper should be published after some points have been clarified by the authors:
1. line 34: "Across multiple ..." - is this sentence complete?
2. Table 4a: Is there a difference between "JB" and "Jb"?
3. line 431: "CI" has not been defined.
4. line 498: "... may shorten the instrument for the fluid and tapping modes ..." - what to the authors mean with that sentence? Shorten the instrument, or shorten the scan time?
5. On page 10, the authors start to describe image resolution and how it is affected by various parameters. Possible contamination of the tips, which might not be reproducible, is not satisfactorily discussed. How often have tips been changed for the images used in this study?
Author Response
Dear Reviewer,
The authors would like to thank you for your valuable comments.
They have been individually addressed in the document attached.
